# Meta-Gradient Reinforcement Learning with an Objective Discovered Online

**Zhongwen Xu, Hado van Hasselt, Matteo Hessel**
**Junhyuk Oh, Satinder Singh, David Silver**
DeepMind
{zhongwen,hado,mtthss,junhyuk,baveja,davidsilver}@google.com

## Abstract

Deep reinforcement learning includes a broad family of algorithms that parameterise an internal representation, such as a value function or policy, by a deep neural network. Each algorithm optimises its parameters with respect to an objective, such as Q-learning or policy gradient, that defines its semantics. In this work, we propose an algorithm based on meta-gradient descent that discovers its own objective, flexibly parameterised by a deep neural network, solely from interactive experience with its environment. Over time, this allows the agent to learn how to learn increasingly effectively. Furthermore, because the objective is discovered online, it can adapt to changes over time. We demonstrate that the algorithm discovers how to address several important issues in RL, such as bootstrapping, non-stationarity, and off-policy learning. On the Atari Learning Environment, the meta-gradient algorithm adapts over time to learn with greater efficiency, eventually outperforming the median score of a strong actor-critic baseline.

## 1   Introduction

Recent advances in supervised and unsupervised learning have been driven by a transition from handcrafted expert features to deep representations [15]; these are typically learned by gradient descent on a suitable objective function to adjust a rich parametric function approximator. As a field, reinforcement learning (RL) has also largely embraced the transition from handcrafting features to handcrafting objectives: deep function approximation has been successfully combined with ideas such as TD-learning [30, 34], Q-learning [42, 23], double Q-learning [36, 37], n-step updates [32, 14], general value functions [33, 18], distributional value functions [7, 3], policy gradients [43, 21] and a variety of off-policy actor-critics [8, 10, 29]. In RL, the agent doesn't have access a differentiable performance metric, thus choosing the right proxy is of particular importance: indeed, each of the aforementioned algorithms differs fundamentally in their choice of objective, designed in each case by expert human knowledge. The deep RL version of these algorithms is otherwise very similar in essence: updating parameters via gradient descent on the corresponding objective function.

Our goal is an algorithm that instead *learns* its own objective, and hence its own deep reinforcement learning algorithm, solely from experience of interacting with its environment. Following the principles of deep learning, we parameterise the objective function by a rich function approximator, and update it by *meta-gradient* learning [28, 1, 11, 44, 47, 39, 2, 20] – i.e. by gradient descent on the sequence of gradient descent updates resulting from the choice of objective function – so as to maximise a naive outer loss function (such as REINFORCE) with minimal initial knowledge.

Importantly, and in contrast to the majority of recent work on meta-learning [11, 2, 20], our meta-gradient algorithm learns **online**, on a single task, during a single "lifetime" of training. This online approach to meta-learning confers several advantages. First, an online learning algorithm can be applied to any RL environment, and does not require a distribution of related environments, nor the

ability to reset and rerun on different environments. Second, an online learning algorithm can adapt the objective function as learning progresses, rather than assume a global, static "one-size-fits-all" objective. Our hypothesis is that an online meta-gradient learning agent will, over time, learn to learn with greater efficiency, and in the long-run this will outperform a fixed (handcrafted) objective.

We show in toy problems that our approach can discover how to address important issues in RL, such as bootstrapping and non-stationarity. We also applied our algorithm for online discovery of an off-policy learning objective to independent training runs on each of 57 classic Atari games. Augmented with a simple heuristic to encourage consistent predictions, our meta-gradient algorithm outperformed the median score of a strong actor-critic baseline on this benchmark.

## 2   Related Work

The idea of learning to learn by gradient descent has a long history. In supervised learning, IDBD and SMD [31, 28] used a meta-gradient approach to adapt the learning rate online so as to optimise future performance. "Learning by gradient descent to learn by gradient descent" [1] used meta-gradients, offline and over multiple lifetimes, to learn a gradient-based optimiser, parameterised by a "black-box" neural network. MAML [11] and REPTILE [24] also use meta-gradients, offline and over multiple lifetimes, to learn initial parameters that can be optimised more efficiently.

In reinforcement learning, methods such as *meta reinforcement learning* [40] and $RL^2$ [9] allow a recurrent network to jointly represent, in its activations, both the agent's representation of state and also its internal parameters. Xu et al [44] introduced metagradients as a general but efficient approach for optimising the meta-parameters of gradient-based RL agents. This approach has since been applied to many different meta-parameters of RL algorithms, such as the discount $\gamma$ and bootstrapping parameter $\lambda$ [44], intrinsic rewards [47, 46], auxiliary tasks [39], off-policy corrections [45], and to parameterise returns as a linear combination of rewards [41] (without any bootstrapping). The metagradient approach has also been applied, offline and over multiple lifetimes, to black-box parameterisations, via deep neural networks, of the entire RL algorithm [2, 20] and [25] (contemporaneous work); evolutionary approaches have also been applied [17].

No prior work has addressed the most ambitious combination: a black-box approach that can parameterise the RL algorithm, meta-learned online, during the course of a single agent lifetime.

The table below catalogs related work on meta-gradient RL. We differentiate methods by several key properties. First, whether they are single lifetime (i.e. they learn online to improve performance by interacting with an environment), or require multiple lifetimes (i.e. they improve performance across repeated agent "lifetimes", each of which faces a different environment sampled from a suitable distribution). Second, whether they are white-box methods (i.e. they meta-learn the hyper-parameters of an existing RL update rule) or black-box methods (i.e. they meta-learn a general-purpose neural network encoding an RL update rule). Third, whether they compute meta-gradients by forward-mode or backward-mode differentiation (or do not use meta-gradients at all). Finally, what is meta-learned.

| Algorithm | Algorithm properties | | | What is meta-learned? |
|---|---|---|---|---|
| IDBD, SMD [31, 28] | † | □ | → | learning rate |
| $SGD^2$ [1] | ††† | ■ | ← | optimiser |
| $RL^2$, Meta-RL [9, 40] | ††† | ■ | X | recurrent network |
| MAML, REPTILE [11, 24] | ††† | □ | ← | initial params |
| Meta-Gradient [44, 47] | † | □ | → | $\gamma$, $\lambda$, reward |
| Meta-Gradient [39, 45, 41] | † | □ | ← | auxiliary tasks, hyperparams, reward weights |
| $ML^3$, MetaGenRL [2, 20] | ††† | ■ | ← | loss function |
| Evolved PG [17] | ††† | ■ | X | loss function |
| Oh et al. 2020 [25] | ††† | ■ | ← | target vector |
| This paper | † | ■ | ← | target |

□ white box, ■ black box, † single lifetime, ††† multi-lifetime

← backward mode, → forward mode, X no meta-gradient

# 3  Algorithm

In this section, we describe our proposed algorithm for online learning of reinforcement learning objectives using meta-gradients. Our starting point is the observation that a single quantity, the update target $G$, plays a pivotal role in characterising the objective of most deep RL algorithms; therefore, the update target offers an ideal entry point to flexibly parameterise the overall RL algorithm.

We first review the objectives used in *classic* RL algorithms and discuss how they may be flexibly parameterised via a neural network in order to expose them to learning instead of being manually designed by human researchers. We then recap the overall idea of meta-gradient reinforcement learning, and we illustrate how it can be used to meta-learn, online, the neural network that parametrises the update target. We discuss this for prediction, value-based control and actor-critic algorithms.

## 3.1  Update Targets in Reinforcement Learning Algorithms

To learn a value function $v_\theta(S)$ in temporal difference (TD) prediction algorithms, in each state we use bootstrapping to construct a *target* $G$ for the value function. For a trajectory $\tau_t = \{S_t, A_t, R_{t+1}, \dots\}$, the target $G_t$ for the one-step TD algorithm is

$$G_t = R_{t+1} + \gamma v_\theta(S_{t+1}); \tag{1}$$

with stochastic gradient descent, we can then update the parameter $\theta$ of the value function as follows:

$$\theta \leftarrow \theta + \alpha(G_t - v_\theta(S_t))\nabla_\theta v_\theta(S_t), \tag{2}$$

where $\alpha$ is the learning rate used to update the parameter $\theta$.

Similarly in value-based algorithms for control, we can construct the update target for the *action-values* $q_\theta(S_t, A_t)$; for instance, in one-step $Q$-learning parameters $\theta$ for the action-value function $q_\theta$ can be updated by:

$$\theta \leftarrow \theta + \alpha(G_t - q_\theta(S_t, A_t))\nabla_\theta q_\theta(S_t, A_t) \quad \text{where} \quad G_t = R_{t+1} + \gamma \max_a q_\theta(S_{t+1}, a). \tag{3}$$

More generally, $n$-step update targets can bootstrap from the value estimation after accumulating rewards for $n$ steps, instead of just considering the immediate reward. For instance in the case of prediction, we may consider the $n$-step truncated and bootstrapped return defined as:

$$G_t = R_{t+1} + \gamma R_{t+2} + \gamma^2 R_{t+3} + \cdots + \gamma^n v_\theta(S_{t+n}). \tag{4}$$

## 3.2  Parameterising RL objectives

In this work, instead of using a discounted cumulative return, we fully parameterise the update target by a neural network. This *meta-network* takes the trajectory as input and produces a scalar update target, i.e., $G = g_\eta(\tau_t)$, where the function $g_\eta : \tau_t \to \mathbb{R}$ is a neural network with parameters $\eta$. We train the meta-network using an end-to-end meta-gradient algorithm, so as to learn an update target that leads to good subsequent performance.

A different way to learn the RL objective is to directly parameterise a *loss* by a meta-network [2, 20], rather than the *target* of a loss. For instance, a standard TD learning update can either be represented by a TD loss $g_\eta(\tau) = (R_{t+1} + \gamma \perp (v_\theta(S_{t+1})) - v_\theta(S_t))^2$, or by a squared loss with respect to a TD target, $g_\eta(\tau) = R_{t+1} + \gamma \perp (v_\theta(S_{t+1}))$, where $\perp$ represents a gradient-stopping operation. Both forms of representation are rich enough to include a rich variety of reinforcement learning objectives.

However, we hypothesise that learning a target will lead to more stable online meta-gradients than learning a loss. This is because the induced learning rule is inherently moving *towards* a target, rather than potentially away from it, thereby reducing the chance of immediate divergence. Because we are operating in an online meta-learning regime, avoiding divergence is of critical importance. This contrasts to prior work in offline meta-learning [2, 20], where divergence may be corrected in a subsequent lifetime.

## 3.3  Meta-Gradient Reinforcement Learning

Meta-gradient reinforcement learning is a family of gradient-based meta learning algorithms for learning and adapting differentiable components (denoted as meta-parameters $\eta$) in the RL update

rule. The key insight of meta-gradient RL is that most commonly used update rules are differentiable, and thus the effect of a sequence of updates is also differentiable.

Meta-gradient RL is a two-level optimisation process. A meta-learned inner loss $L_\eta^{\text{inner}}$ is parameterised by meta-parameters $\eta$ and the agent tries to optimise $L_\eta^{\text{inner}}$ to update $\theta$. Given a sequence of trajectories $\mathcal{T} = \{\tau_i, \tau_{i+1}, \tau_{i+2}, \ldots, \tau_{i+M}, \tau_{i+M+1}\}$, we apply multiple steps of gradient descent updates to the agent $\theta$ according to the inner losses $L_\eta^{\text{inner}}(\tau_i, \theta_i)$. For each trajectory $\tau \in \mathcal{T}$, we have:

$$\Delta\theta_i \propto \nabla_{\theta_i} L_\eta^{\text{inner}}(\tau_i, \theta_i) \qquad \theta_{i+1} = \theta_i + \Delta\theta_i . \tag{5}$$

Consider keeping $\eta$ fixed for $M$ updates to the agent parameter $\theta$:

$$\theta_i \xrightarrow{\eta} \theta_{i+1} \xrightarrow{\eta} \ldots \xrightarrow{\eta} \theta_{i+M-1} \xrightarrow{\eta} \theta_{i+M} . \tag{6}$$

A differentiable outer loss $L^{\text{outer}}(\tau_{i+M+1}, \theta_{i+M})$ is then applied to the updated agent parameters $\theta' = \theta_{i+M}$. The gradient of this loss is taken w.r.t. meta-parameters $\eta$ and then used to update $\eta$ via gradient descent:

$$\Delta\eta \propto \nabla_\eta L^{\text{outer}}(\tau_{i+M+1}, \theta_{i+M}) \qquad \eta \leftarrow \eta + \Delta\eta . \tag{7}$$

We call this quantity $\nabla_\eta L_\eta^{\text{outer}}$ the meta-gradient. We can iterate this procedure during the training of the agent $\theta$ and repeatedly update the meta-parameters $\eta$.

The meta-gradient flows through the multiple gradient descent updates to the agent $\theta$, i.e., the whole update procedure from $\theta_i$ to the outer loss of the final updated agent parameter $\theta_{i+M}$. By applying chain rule, we have $\frac{\partial L^{\text{outer}}}{\partial \eta} = \frac{\partial L^{\text{outer}}}{\partial \theta'} \frac{\partial \theta'}{\partial \eta}$. In practice, we can use automatic differentiation packages to compute the meta gradient $\frac{\partial L^{\text{outer}}}{\partial \eta}$ with comparable compute complexity as the forward pass.

The meta-gradient algorithm above can be applied to any differentiable component of the update rule, for example to learn the discount factor $\gamma$ and bootstrapping factor $\lambda$ [44], intrinsic rewards [47, 46], and auxiliary tasks [39]. In this paper, we apply meta-gradients to learn the meta-parameters of the update target $g_\eta$ online, where $\eta$ are the parameters of a neural network. We call this algorithm FRODO (Flexible Reinforcement Objective Discovered Online). The following sections instantiate the FRODO algorithm for value prediction, value-based control and actor-critic control, respectively.

## 3.4   Learned Update Target for Prediction and Value-based Control

Given a learned update target $g_\eta$, we propose updating the predictions $v_\theta$ towards the target $g_\eta$. With a squared loss, this results in the update

$$\Delta\theta \propto (g_\eta(\tau) - v_\theta(S))\nabla_\theta v_\theta(S), \tag{8}$$

where the meta-network parameterised by $\eta$ takes the trajectory $\tau$ as input and outputs a scalar $g_\eta(\tau)$. After $M$ updates, we compute the outer loss $L^{\text{outer}}$ from a validation trajectory $\tau'$ as the squared difference between the predicted value and a canonical multi-step bootstrapped return $G(\tau')$, as used in classic RL:

$$\nabla_{\theta'} L^{\text{outer}} = (G(\tau') - v_{\theta'}(S'))\nabla_{\theta'} v_{\theta'}(S') \tag{9}$$

can then be plugged in Equation (7), to update $\eta$ and continue with the next iteration. Here $\theta_t$ is interpreted and treated as a function of $\eta$, which was held fixed during several updates to $\theta$.

For value-based algorithms in control, the inner update is similar, but the learned target is used to update an action-value function $q_\theta(S, A)$ in the inner update. Any standard RL update can be used in the outer loss, such as Q-learning [42], Q($\lambda$), or (Expected) Sarsa [27, 38].

## 3.5   Learned Update Target for Actor-Critic Algorithms in Control

In actor-critic algorithms, the update target is used both to compute policy gradient update to the policy, as well as to update the critic. We form an A2C [21] update with $g_\eta(\tau)$:

$$\Delta\theta \propto (g_\eta(\tau) - V(S))\nabla_\theta \log \pi(S, A) + c_1(g_\eta(\tau) - V(S))\nabla_\theta v(S) + c_2 \nabla_\theta H(\pi(S)), \tag{10}$$

where $H(\cdot)$ denotes the entropy of the agent's policy, and $c_1$ and $c_2$ denote the weightings of the critic loss and entropy regularisation terms, respectively.

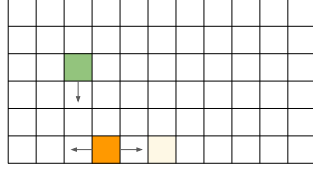

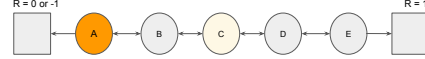

(b) "5-state random walk" for non-stationarity

(a) "Catch" for learning bootstrapping

Figure 1: Illustrations of toy environments used in motivating examples: (a) An instance of the $6 \times 11$ "Catch" [22] control environment. The agent controls the paddle on the bottom row (in orange), which starts the episode in the centre (faint orange). The pellet following from the top is depicted in green. (b) An instance the 5-state Random Walk prediction environment. The agent starts in state $C$ and moves randomly to the left or the right. The squares denote terminal states. The leftmost termination reward is non-stationary.

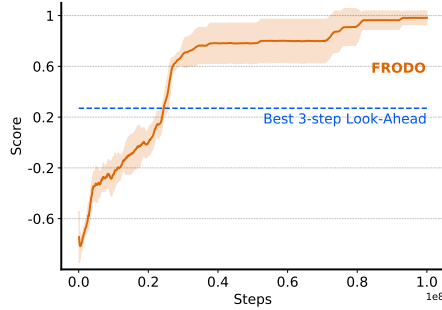

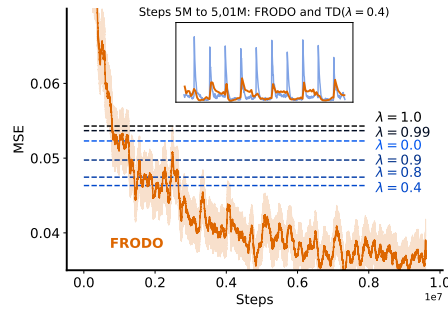

(a) Performance on "Catch"

(b) Performance on "5-state Random Walk"

Figure 2: Performance on the toy problems: (a) The mean episode return on "Catch". The solid orange line corresponds to FRODO. The blue dashed line denotes the best performance for a three step look ahead agent, performance above that line implies some form of bootstrapping has been learned. (b) The mean squared error in "5-state Random Walk" between the learned value function and the true expected returns, averaged across all states. The solid orange line denotes the performance achieved by FRODO over the course of training. The dashed blue lines correspond to the performance of TD($\lambda$), for various $\lambda$s. The inset shows a zoom of the value error around $5M$ steps, for FRODO, in orange, and for the best performing TD($\lambda$), i.e. for $\lambda = 0.4$.

The meta-gradient can be computed on the validation trajectory $\tau'$ using the classic actor-critic update:

$$\nabla_{\theta'} L^{\text{outer}} = (G(\tau') - V(S'))\nabla_{\theta'} \log \pi(S', A') + c_1 (G(\tau') - V(S'))\nabla_{\theta'} v(S') + c_2 \nabla_{\theta'} H(\pi(S')), \tag{11}$$

where $\theta'$ is the updated agent after $M$ updates to the agent parameter $\theta$. According to the chain rule of meta-gradient, we obtain the gradient of $\eta$ and update $\eta$ accordingly. Note that one can use either $n$-step return (including $\lambda$-return [32]) as $G(\tau)$, or to use VTrace-return [10] to enable off-policy correction.

## 4 Motivating Examples

In this section, we explore the capability of FRODO to discover how to address fundamental issues in RL, such as bootstrapping and non-stationarity, based on simple toy domains. Larger scale experiments will subsequently be presented addressing off-policy learning.

**Bootstrapping**: We use a simple $6 \times 11$ environment called "Catch" [22]. The agent controls a paddle located on the bottom row of the grid. The agent starts in the centre and, on each step, it can move on cell to the left, one cell to the right or stand still. At the beginning of each episode a pellet appears in a random start location on the top row. On each step, the pellet move down on cell. When the pellet reaches the top row the episode terminates. The agent receives a reward of 1 if it caught the pellet, -1 otherwise. All other rewards are zero. See Figure 1a, for a depiction of the environment.

We applied FRODO to learning to control the agent. We used the the *full Monte Carlo return* as update target in the outer loss. In the inner update, instead, the agent only received a trajectory with

3 transitions. This requires FRODO to discover the concepts of temporal-difference prediction and bootstrapping – learning to estimate and use a prediction about events outside of the data – since the game cannot be solved perfectly by looking ahead just three steps. We conduct 10 independent runs with random seeds. From Figure 2a, in orange, we report the average episode return of FRODO observed during the course of training. The policy learned by FRODO surpassed the best possible performance for a 3 step look-ahead agent (the dashed blue line), and learned to control the agent optimally (an average episode return of 1).

**Non-Stationarity**: We use a non-stationary variant of the "5-state Random Walk" environment [32]. The agent starts in the centre of a chain, depicted on Figure 1b, and moves randomly left or right on each step. Episodes terminate when the agent steps out of either boundary, on termination a reward is provided to the agent; on the left side, the reward is either $0$ or $-1$ (switching every 960 time-steps, which corresponds to 10 iterations of FRODO), on the right side the reward is always 1. Each trajectory has 16 time steps.

We applied FRODO to predict the value function, using a TD(1) update as an outer loss. The critical issue is the non-stationarity; the agent must quickly adapt its prediction whenever the reward on the leftmost side switched from 0 to 1, or vice versa. FRODO learned an update capable of dealing with such non-stationarity effectively. In the experiment, we perform 10 independent runs. In Figure 2b, we report the mean squared error of the predictions learned by FRODO, in orange. The dashed horizontal lines correspond to the average error of the predictions learned by TD($\lambda$) at convergence. The update learned online by FRODO resulted in more accurate predictions, compared to those learned by the TD($\lambda$) update, for any value of $\lambda$. The plot zooms into the period around $5M$ steps for FRODO (in orange) and for the best performing TD($\lambda$), i.e. $\lambda = 0.4$; the predictions learned by FRODO adapted much more robustly to change-points than those of TD($\lambda$), as demonstrated by the significantly lower spikes in the prediction error.

## 5 Large-Scale Experiments

In this section we scale up the actor-critic variant of FRODO from Section 3.5 to more complex RL environments in the Arcade Learning Environment. We instantiate our algorithm within a distributed framework, based on an actor-learner decomposition [10], and implemented in JAX [5]. The implementation details, computing infrastructure and pseudo-code are provided in Appendix.

### 5.1 Off-Policy Learning

In actor-learner architectures [10], a parallel group of actors collect trajectories by interacting with separate copies of the environment, and send the trajectories to a queue for the learner to process in batch. This architecture enables excellent throughput, but the latency in communications between the actors and the learner introduces off-policy learning issues, because the parameters of the actors' policy (the behaviour policy $\mu$) lag behind the parameters in the learner's policy (the target policy $\pi$). Actor-critic updates such as VTrace can correct for such off-policyness by using the action probabilities under the behaviour policy to perform importance sampling.

To address this in our actor-critic instantiation of FRODO, we use VTrace [10], rather than a vanilla actor-critic, as outer update. In the inner loop, the meta network takes trajectories from the behaviour policy $\mu$ as input. Specifically it receives the rewards $R_t$, discounts $\gamma_t$, and, as in the motivating examples from Section 4, the values from future time-steps $v(S_{t+1})$, to allow bootstrapping from the learned predictions. To address off-policy learning, the probabilities of the target policy and behaviour policy for the current action $A_t$ selected in state $S_t$, (i.e., $\pi(S_t, A_t)$ and $\mu(S_t, A_t)$), are also fed as inputs. This allows the inner loss to potentially discover off-policy algorithms, by constructing suitable off-policy update targets for the policy and value function. Thus, inputs to the meta network include $\{R_{t+1}, \gamma_{t+1}, v(S_{t+1}), \pi(S_t, A_t), \mu(S_t, A_t), \cdots\}$. The meta network is parameterised by an LSTM [16] and processes each trajectory in reverse order to compute the target $G_\eta(\tau)$.

### 5.2 Consistent Prediction

In our large scale experiments, we consider a simple yet effective heuristic which enables dramatic speed ups of learning in complex environments. While the meta-networks have the expressive power to model any target function that takes a trajectory as input, we regularise the learning space of the

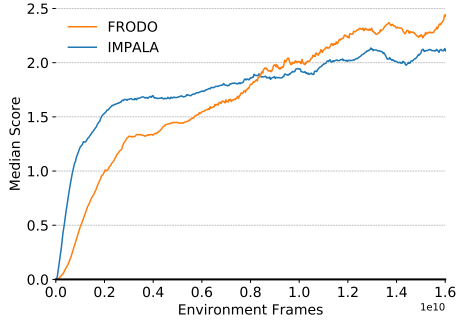

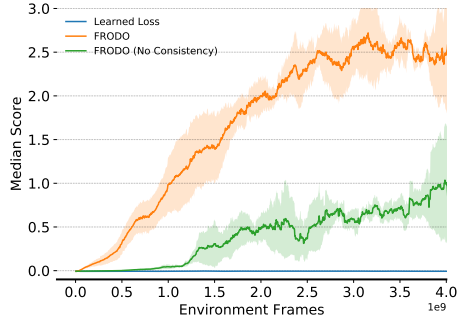

(a) Comparison between FRODO and an IMPALA baseline, in terms of the median human-normalised score across 57 Atari games. FRODO takes longer to take-off but eventually outperforms IMPALA.

(b) Comparison (on 8 games) of several metagradient algorithms, where the meta-network either parametrises the loss (in blue), or the target, with (in orange) and without (in green) regularisation.

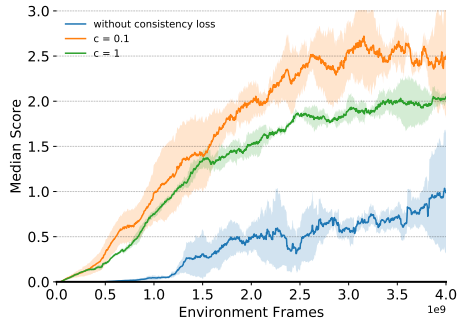

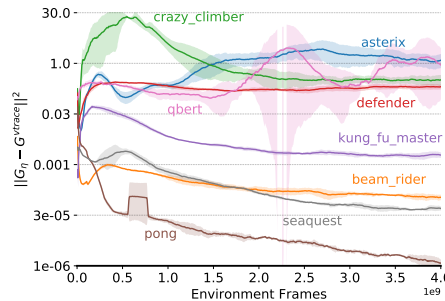

(c) A parameter study for the value of the consistency weight $c$. Too high value of this hyperparameter can reduce performance in the long run.

(d) Difference between the vtrace target and the learned update target $G_\eta$. We observe large variations throughout training and across games.

Figure 3: Figure (a) shows the experimental results on the full set of 57 Atari games. Figures (b), (c), and (d) show additional ablations and analysis on a selection of 8 Atari games: asterix, beam rider, crazy climber, defender, kung fu master, pong, qbert, and seaquest. In all cases the x-axis reports the total number of environment frames, for each game. See y-axis differs between subfigures.

target function towards targets that are self-*consistent* over time (a property that is common to most update targets in deep reinforcement learning - c.f. [26]). Concretely, we suggest to regularise the learned update targets $G_\eta$ towards functions that decompose as:

$$G_t^\eta = R_{t+1} + \gamma G_{t+1}^\eta. \tag{12}$$

To incorporate the heuristic into our meta-gradient framework, we transform the above equations into a prediction loss, and add this component into $L^{\text{outer}}$ to learn our meta-network $\eta$. For example, using a one-step prediction consistency loss:

$$L^{\text{outer}} \leftarrow L^{\text{outer}} + c||\perp(R_{t+1} + \gamma G_{t+1}^\eta) - G_t^\eta||^2, \tag{13}$$

where $c$ is for a coefficient for the consistency loss and $\perp$ denotes stop gradient operator. Extension to $n$-step self-consistent prediction can be obtained by decomposing $G_t^\eta$ into $n$-step cumulative discounted rewards with bootstrapping in the final step.

## 5.3 Atari Experiments

We evaluated the performance of our method on a challenging and diverse set of classic Atari games, from the Arcade Learning Environment (ALE) [4].

We applied the FRODO algorithm to learn a target online, using an outer loss based on the actor-critic algorithm IMPALA [10], and using a consistency loss was included with $c = 0.1$. The agent network is parameterised with a two-layer convolutional neural network (detailed configurations

and hyperparameters can be found in the Appendix). We evaluate our agents over 16 billion frames of training. We ran separate training runs for all 57 standard benchmark games in the ALE. We computed the median of human-normalised scores throughout training, and compared to the same IMPALA actor-critic algorithm without any objective discovery. Note that FRODO algorithm does introduce algorithmic complexity compared to the IMPALA baseline, thus we provide pseudo-code in Appendix C to facilitate understanding and reproducibility.

In Figure 3a we see that the meta-gradient algorithm learned slowly and gradually to discover an effective objective. However, over time the meta-gradient algorithm learned to learn more rapidly, ultimately overtaking the actor-critic baseline and achieving significantly stronger final results. We hypothesise the performance advantage is led by the adaptive nature of the learned objective, which allows the agent to find most suitable objective according to its learning context along the way, instead of using a traditional global static objective function.

## 5.4 Analysis

We now examine the technical contributions that facilitate our primary goal of online objective discovery: representing targets in the meta-network versus representing a loss; and the introduction of a consistency loss. In these analysis experiments, we use a subset of 8 Atari games, namely, "kung fu master", "qbert", "crazy climber", "asterix", "beam rider", "defender", "pong" & "seaquest", and train each of the games over three independent runs. In each of the plots, we show the *median* human-normalised score over all three runs; the shaded area shows standard derivation across random seeds. Ablation runs were performed for 4 billion frames.

**Discovering targets v.s. Discovering loss**: Our first experiment compares the performance of online objective discovery between a meta-network that represents the *target* and a meta-network that represents the *loss*, similarly to prior work in offline, multi-lifetime setups such as ML$^3$ [2], MetaGenRL [20]. As we illustrate in Figure 3b, directly representing the loss by the meta-network performs poorly across all games. We hypothesise that this is due to significant instabilities in the learning dynamics, which may at any time form a loss that leads to rapid divergence. In contrast, representing the target by the meta-network performs much more stably across all games.

**Consistency loss**: Next, we examine the effectiveness of a consistency loss in large-scale experiments. We use values of different magnitude as the coefficient of the consistency loss in FRODO, varying between disabling consistency loss (coefficient $c = 0$) and a large consistency loss ($c = 1$). The aggregated median score learning curves are shown in Figure 3c. The introduction of a modest level ($c = 0.1$) of consistency loss led to a dramatic improvements in learning speed and achieved significantly higher performance. Without the consistency heuristic, performance dropped significantly and was also more unstable, presumably due to an increased likelihood of uninformative or misleading targets. Additionally, regularising too strongly ($c = 1$) led to significantly worse performance.

**Analysis of Learned Objective:** Finally, we analysed the objective learned by FRODO over time. Our primary question was whether the discovered target used in the inner loss differed significantly from the VTrace target used in the outer loss. For each of the eight games, we computed the mean-squared error, averaged over the time-steps in the trajectory, between the VTrace return and the meta-network return $g_\eta(\tau)$. Figure 3d shows that the discovered objective both varies over time, and varies significantly away from the VTrace target, with a very different characteristic in each game. Only in the game of "Pong" was a target close to VTrace preferred throughout training, perhaps because nothing more complex was required in this case.

## 6   Conclusion

In this paper, we proposed an algorithm that allows RL agents to learn their own objective during online interactions with their environment. The objective, specifically the target used to update the policy and value function, is parameterised by a deep neural meta-network. The nature of the meta-network, and hence the objective of the RL algorithm, is discovered by meta-gradient descent over the sequence of updates based upon the discovered target.

Our results in toy domains demonstrate that FRODO can successfully discover how to address key issues in RL, such as bootstrapping and non-stationarity, through online adaptation of its objective. Our results in Atari demonstrate that FRODO can successfully discover and adapt off-policy learning

objectives that are distinct from, and performed better than, strong benchmark RL algorithms. Taken together, these examples illustrate the generality of the proposed method, and suggest its potential both to recover existing concepts, and to discover new concepts for RL algorithms.

## Broader Impact

This work is situated within a broad and important long-term research program for reinforcement learning: how might a machine discover its own algorithm for RL? Specifically, the research considers one strand of potential research in this area, which is how a machine might discover its own objective for RL. Because our research focuses on the online setting (compared, for example, to much prior work on meta-learning that learns offline from a distribution of tasks), it is applicable to any RL problem. In principle, therefore, any benefits demonstrated in this paper might potentially be applicable to other future RL problems. Thus, the ethical consequences of this research are similar to those for any other research on the RL problem itself: it may provide some progress and accelerate research towards all RL problems, which may benefit any users and use-cases of RL (both "good" and "bad"). Our algorithm learns entirely from interaction with its environment, and does not utilise any external source of data.

## Acknowledgments and Disclosure of Funding

The authors would like to thank Manuel Kroiss, Iurii Kemaev and developers of JAX, Haiku, RLax, Optax for their kind engineering support; and thank Joseph Modayil, Doina Precup for their comments and suggestions on the paper.

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
