[Supplementary Material]

# A Implementation Details

In this section, we provide the implementation details of our proposed algorithm, including the motivating examples we described in Sec 4 and large-scale experiments in Atari in Sec 5.

## A.1 Motivating Examples

In our investigations, a wide range of hyperparameters work well in the motivating examples and we provide the specific one we used to show our results.

**Bootstrapping:** For the bootstrapping experiment, we use an LSTM [16] with 256 hidden units as meta network. The inputs to the meta network include reward $R_{t+1}$, discount $\gamma_{t+1}$, value from the next state $v(S_{t+1})$, which were fed in reverse-time order into the LSTM. The agent network is a two layer MLP with 256 hidden units. Both agent optimiser and the meta optimiser are RMSProp [35], where learning rate is $1e-3$ and meta learning rate is $1e-4$. $M = 5$ inner updates are performed to the agent then the metagradient is obtained. We perform 10 independent runs with 10 different random seeds and report average performance with standard deviation.

**Non-stationarity:** For the non-stationarity experiment, we use an LSTM with 32 hidden units as meta network. Each trajectory from the environment has 16 transitions. The inputs to the meta network include reward $R_{t+1}$, discount $\gamma_{t+1}$, value from the next state $v(S_{t+1})$, which were fed in reverse-time order into the LSTM. Both agent optimiser and the meta optimiser are RMSProp [35], where learning rate is $1e-1$ and meta learning rate is $1e-2$. We use larger learning rates here to enable rapid learning in the simple random walk environment. Same as above, $M = 5$ inner updates are performed to the agent then the metagradient is obtained. For our baseline TD($\lambda$) experiment, we use the same learning rate $1e-1$ and report convergent result from all $\lambda$s from [ 0., 0.4, 0.8, 0.9, 0.99, 1 ]. We perform 10 independent runs with 10 different random seeds and report average performance with standard deviation.

## A.2 Large-Scale Experiments

In the large-scale Atari experiments, we compare FRODO with the corresponding baseline IM-PALA [10], where we keep all the hyper-parameters on the agent learning the same in the two agents. In Table 1, we show the shared hyper-parameter between FRODO and IMPALA agents in "IMPALA hyper-parameter", which includes network architecture, batch size, learning rate, etc, and we show the FRODO specific hyper-parameter in the bottom, such as inner update steps, meta learning rate, etc.

# B Computing Infrastructure

We run our experiments using a distributed infrastructure implemented in JAX [5]. The computing infrastructure is based on an actor-learner decomposition [10], where multiple actors generate experience in parallel, and this experience is channelled into a learner via a small queue. Both the actors and learners are co-located on a single machine, where the host is equipped with 56 CPU cores and connected to 8 TPU cores [19]. To minimise the effect of Python's Global Interpreter Lock, each actor-thread interacts with a *batched environment*; this is exposed to Python as a single special environment that takes a batch of actions and returns a batch of observations, but that behind the scenes steps each environment in the batch in C++. The actor threads share 2 of the 8 TPU cores (to perform inference on the network), and send batches of fixed size trajectories of length T to a queue. The learner threads takes these batches of trajectories and splits them across the remaining 6 TPU cores for computing the parameter update (these are averaged with an all reduce across the participating cores). Updated parameters are sent to the actor TPU devices via a fast device to device channel as soon as the new parameters are available. This minimal unit can be replicates across multiple hosts, each connected to its own 56 CPU cores and 8 TPU cores, in which case the learner updates are synced and averaged across all learner cores (again via fast device to device communication).

It took approximately 1.5 hours to learn from 1 billion Atari environment frames with the two-layer ConvNet we used.

| IMPALA hyper-parameter | Value |
|---|---|
| Network architecture | Two-layer ConvNet |
| ConvNet channels | 32, 64 |
| ConvNet kernel sizes | 4, 4 |
| ConvNet kernel strides | 4, 4 |
| Unroll length | 30 |
| Batch size | 30 |
| Baseline loss scaling | 0.5 |
| Entropy cost | 0.01 |
| Learning rate | $1e{-}3$ |
| RMSProp momentum | 0.0 |
| RMSProp decay | 0.99 |
| RMSProp $\epsilon$ | 0.1 |
| Clip global gradient norm | 1e4 |
| **FRODO hyper-parameter** | **Value** |
| Inner update steps $M$ | 5 |
| Meta learning rate | $5e{-}4$ |
| Meta optimiser | RMSProp |
| Clip Meta gradient norm | 1e4 |
| Meta batch size | 30 |

Table 1: Detailed hyper-parameters for Atari experiments.

## C  Pseudo-code

In this section, we provide pseudo-code in Python format which show how we implemented the key ideas in our proposed FRODO algorithm. Our algorithm is implemented with neural network library Haiku [12], reinforcement learning library RLax [6] and optimiser library Optax [13] in JAX [5]. We first use the prediction task of toy problems in Sec 4 as an example to show how we implemented the algorithm by a few small functions. Then, we show a base class for FRODO agents and concrete instantiation for our Atari experiments. LICENCE shown in Sec C.1 applies to all the provided code.

### C.1  LICENCE

### C.2  Prediction task in toy problems

In the prediction task of non-stationary "5-state random walk" (Sec 4), the meta network $g_\eta$ takes a trajectory $\tau$ as input and outputs $G_\eta(\tau)$ as the update target (shown in `outer_loss_fn`), then the agent use squared TD error between $G_\eta(\tau)$ and the values to update the value function (shown in `agent_update`). The agent applies a few steps of updates then obtains a differentiable outer loss (shown in agent_updates_and_outer_loss). Finally we have the `two_level_update` to update the meta network $\eta$ and complete the algorithm.

```python
def inner_loss_fn(theta, eta, traj):
  """Inner loss function."""
  obs = traj.observation
```

```python
    rewards = traj.reward[1:]
    discounts = traj.discount[1:]
    v = v_net_apply(theta, obs)

    eta_inputs = {
        'obs': obs[:-1],
        'reward': rewards,
        'discount': discounts,
        'v': v[1:],
    }
    g_eta = eta_net_apply(eta, eta_inputs)
    td_error = g_eta - v[:-1]
    return jnp.mean(jnp.square(td_error))

def outer_loss_fn(theta, traj):
    """Outer loss function."""
    obs = traj.observation
    rewards = traj.reward[1:]
    discounts = traj.discount[1:]
    v = v_net_apply(theta, obs)
    g = discounted_return_fn(rewards, discounts, v[-1, :][None, ...])
    td_error = jax.lax.stop_gradient(g) - v[:-1]
    return jnp.mean(jnp.square(td_error))

def agent_update(theta, opt_state, eta, traj, optim):
    """Update the agent parameter."""
    loss, g = jax.value_and_grad(inner_loss_fn)(theta, eta, traj)
    delta, opt_state = optim.update(g, opt_state)
    theta = optix.apply_updates(theta, delta)
    return theta, opt_state, loss

def agent_updates_and_outer_loss(eta, theta, opt_state, traj_arr, optim):
    # M steps of inner updates to the agent
    for traj in traj_arr[:-1]:
        theta, opt_state, unused_loss = agent_update(theta, opt_state, eta, traj,
                                                     optim)
    val_traj = traj_arr[-1]
    # Take outer loss from the updated theta and validation trajectory
    outer_loss = outer_loss_fn(theta, val_traj)
    return outer_loss, (theta, opt_state)

def two_level_update(theta, opt_state, eta, eta_opt_state, traj_arr, optim,
                     eta_optim):
    # Obtain the meta gradient and perform two-level update
    (loss, (theta, opt_state)), meta_g = jax.value_and_grad(
        agent_updates_and_outer_loss, has_aux=True)(eta, theta, opt_state,
                                                    traj_arr, optim)
    # Update eta with metagradient
    delta_eta, eta_opt_state = eta_optim.update(meta_g, eta_opt_state)
    eta = optix.apply_updates(eta, delta_eta)
    return theta, opt_state, eta, eta_opt_state, loss
```

## C.3 FRODO Base Class

In this section, we provide a `FRODOAgent` base class to show how to implement the agent for large-scale experiments. Similarly to the toy example code above, we use `two_level_update` function as the entry point of the algorithm. To instantiate specific algorithms, we only need to implement the abstract functions `inner_loss` and `outer_loss`, which we are showing an example in C.4 and C.5. Different inner loss and outer loss other than what we used can be instantiated in the subclass.

```python
import jax
from jax import lax
from jax import numpy as jnp
from jax.experimental import optimizers
from jax.tree_util import tree_multimap

class FRODOAgent:
  """Base class for FRODO Agents."""
  def __init__(self, ...):
    """Init function."""
    raise NotImplementedError

  def actor_step(self, params, env_output, actor_state, rng_key, evaluation):
    """Compute actions."""
    raise NotImplementedError

  def build_eta(self, meta_net_type, meta_net_kwargs):
    """Build eta and eta optimizer."""
    raise NotImplementedError

  def inner_loss(self, theta, eta, rollout, first_actor_state):
    """Inner loss on the rollout, the loss contains learnable eta."""
    raise NotImplementedError

  def outer_loss(self, theta, rollout, first_actor_state):
    """Outer loss on the rollout."""
    raise NotImplementedError

  def inner_update(self, theta, theta_opt_state, eta, rollout,
                   first_actor_state):
    """Perform a step of inner update to the agent."""
    # grad(inner_loss)
    g, (last_actor_state, logs) = jax.grad(
        self.inner_loss, has_aux=True)(theta, eta, rollout, first_actor_state)
    # Gather gradients from all devices.
    grads = tree_multimap(
        lambda t: lax.psum(t, axis_name='i') / lax.psum(1., axis_name='i'), g)
    # Clip the gradients according to global max norm.
    grads = optimizers.clip_grads(grads, self.max_theta_grad_norm)
    # Use the optimizer to apply a suitable gradient transformation.
    updates, new_theta_opt_state = self._opt_update(grads, theta_opt_state)
    # Update parameters.
    new_theta = tree_multimap(lambda p, u: p + u, theta, updates)
    return new_theta, new_theta_opt_state, last_actor_state, logs

  def inner_updates_and_outer_loss(self, eta, theta, theta_opt_state,
                                   rollout_arr):
    """Perform a few inner updates and have the outer loss of the last batch."""
    logs_arr = []
    # Perform inner updates on the rollout list.
```

```python
    cons_loss = jnp.array(0.)
    last_actor_states = []
    for _, (rollout, first_actor_state) in enumerate(rollout_arr[:-1]):
      theta, theta_opt_state, last_actor_state, logs = self.inner_update(
          theta, theta_opt_state, eta, rollout, first_actor_state)
      if 'cons_loss' in logs:
        cons_loss += logs['cons_loss']
      logs_arr.append(logs)
      last_actor_states.append(last_actor_state)
    # Use the final batch of rollout as validation data to form the outer loss.
    val_rollout, val_first_actor_state = rollout_arr[-1]
    loss, (last_actor_state, _) = self.outer_loss(theta, val_rollout,
                                                  val_first_actor_state)
    last_actor_states.append(last_actor_state)
    loss = loss + self.settings.cons_weight * cons_loss
    last_actor_state = tree_multimap(lambda *inputs: jnp.concatenate(inputs),
                                    *last_actor_states)
    return loss, (theta, theta_opt_state, last_actor_state, logs)

  def two_level_update(self, theta, theta_opt_state, eta, eta_opt_state,
                       rollout_arr):
    """Two level updates to eta and to theta."""
    # Perform inner updates and compute meta gradient.
    (unused_outer_loss_v, (new_theta, new_theta_opt_state, last_actor_state,
                           logs)), eta_g = jax.value_and_grad(
                               self.inner_updates_and_outer_loss,
                               has_aux=True)(eta, theta, theta_opt_state,
                                             rollout_arr)
    # Gather meta gradients from all devices.
    eta_grads = tree_multimap(
        lambda t: lax.psum(t, axis_name='i') / lax.psum(1., axis_name='i'),
        eta_g)
    eta_grads = optimizers.clip_grads(eta_grads, self.max_eta_grad_norm)
    # Update eta.
    eta_updates, new_eta_opt_state = self._eta_opt_update(
        eta_grads, eta_opt_state)
    new_eta = tree_multimap(lambda p, u: p + u, eta, eta_updates)
    return (new_theta, new_theta_opt_state, new_eta, new_eta_opt_state,
            last_actor_state, logs)
```

## C.4 FRODO Actor-Critic Inner Loss

```python
import rlax

  def inner_loss(self, theta, eta, rollout, first_actor_state):
    env_output, agent_output = self.preprocessing_rollout(rollout)
    actions = agent_output.actions[self._action_name]
    (logits, v, torso_out), last_actor_state = self.net_unroll(
        theta, env_output, env_output['step_type'], first_actor_state)
    logits = logits[self._action_name]
    v_tm1 = jnp.squeeze(v[:-1], axis=-1)
    v_t = jnp.squeeze(v[1:], axis=-1)

    logpi = jax.nn.log_softmax(logits[:-1])  # [T,B,A]
    logpi_a = rlax.batched_index(logpi, actions[:-1])  # [T, B]

    logmu = jax.nn.log_softmax(agent_output.logits[:-1])  # [T,B,A]
    logmu_a = rlax.batched_index(logmu, actions[:-1])  # [T,B]
```

```python
    inputs = dict(
        reward=env_output['reward'][1:],
        discount=env_output['discount'][1:],
        pi_a=jnp.exp(logpi_a),
        mu_a=jnp.exp(logmu_a),
        v_t=v_t,
        torso_out=torso_out[:-1],
    )
    learned_return = self._eta_apply_fn(eta, inputs)
    # Policy gradient loss.
    advantages = learned_return - sg(v_tm1)   # [T, B]
    pi_loss = -jnp.sum(advantages * logpi_a)

    # Baseline loss
    baseline_loss = 0.5 * jnp.sum(jnp.square(learned_return - v_tm1))
    # Entropy loss.
    pi = jax.nn.softmax(logits[:-1])
    logpi = jax.nn.log_softmax(logits[:-1])
    entropy = jnp.sum(-pi * logpi, axis=-1)
    entropy_loss = -jnp.sum(entropy)
    # Sum and weight losses.
    total_loss = pi_loss
    total_loss += self.settings.baseline_weight * baseline_loss
    total_loss += self.settings.entropy_weight * entropy_loss

    # Consistency loss for G_eta
    rewards = env_output['reward'][1:]
    discounts = env_output['discount'][1:]
    learned_return_tm1 = learned_return[:-1]

    # n-step self-consistency.
    batched_discounted_returns = jax.vmap(
        rlax.discounted_returns, in_axes=1, out_axes=1)
    target_tm1 = batched_discounted_returns(rewards, discounts,
                                            learned_return[-1][None, ...])
    cons_loss = 0.5 * jnp.sum(
        jnp.square(sg(target_tm1[:-1]) - learned_return_tm1))

    logs = {'cons_loss': cons_loss}
    return total_loss, (last_actor_state, logs)
```

### C.5 FRODO Actor-Critic Outer Loss

```python
import rlax

  def outer_loss(self, theta, rollout, first_actor_state):
    env_output, agent_output = self.preprocessing_rollout(rollout)
    actions = agent_output.actions[self._action_name]
    # Apply net_unroll.
    (logits, v, _), last_actor_state = self.net_unroll(
        theta, env_output, env_output['step_type'], first_actor_state)
    logits = logits[self._action_name]
    tau = self.settings.temperature
    logits = logits / tau
    v_tm1 = jnp.squeeze(v[:-1], axis=-1)
    v_t = jnp.squeeze(v[1:], axis=-1)
    # Apply `vtrace` computation in batch.
```

```python
batched_vtrace = jax.vmap(
    functools.partial(
        rlax.vtrace_td_error_and_advantage,
        lambda_=self.settings.lambda_,
        clip_rho_threshold=self.settings.clip_rho_threshold,
        clip_pg_rho_threshold=self.settings.clip_pg_rho_threshold),
    in_axes=1, out_axes=1)
# Compute importance weighted td-errors and advantages.
reward = env_output['reward']
discount = env_output['discount']
rhos = rlax.categorical_importance_sampling_ratios(
    logits, agent_output.logits, actions)
vtrace_output = batched_vtrace(
    v_tm1, v_t, reward[1:], discount[1:], rhos[:-1])
# Policy gradient loss.
logpi = jax.nn.log_softmax(logits[:-1])   # [T,B,A]
logpi_a = rlax.batched_index(logpi, actions[:-1])   # [T, B]
advantages = sg(vtrace_output.pg_advantage)  # [T, B]
pi_loss = -jnp.sum(advantages * logpi_a)
# Value loss.
baseline_loss = 0.5 * jnp.sum(jnp.square(vtrace_output.errors))
# Entropy loss.
pi = jax.nn.softmax(logits[:-1])
logpi = jax.nn.log_softmax(logits[:-1])
entropy = jnp.sum(-pi * logpi, axis=-1)
entropy_loss = -jnp.sum(entropy)
# Sum and weight losses.
total_loss = pi_loss
total_loss += self.settings.baseline_weight * baseline_loss
total_loss += self.settings.entropy_weight * entropy_loss
return total_loss, (last_actor_state, None)
```