[Reviews · NeurIPS 2020]

Review 1

Summary and Contributions: This paper proposes an approach for meta-gradient reinforcement learning where the value target used in loss functions is learned in an outer optimization loop. The authors hypothesize that this provides more freedom to an RL agent to learn its own objective for policy optimization, as opposed to priors (such as the Bellman equation) being applied to the learning algorithm. The authors present their algorithm (called FRODO) in both value function and actor critic variants and provide empirical evidence for the effectiveness of their method on toy tasks and the ALE benchmark. UPDATE: Thanks to the authors for their clarifications!

Strengths: * Novel method for meta-RL that is clearly different from prior work (also clearly highlighted in section 2). * Promising empirical results on the tested environments along with useful analyses * Clear exposition and writing

Weaknesses: * Would have liked to see some environments more substantial than Atari (e.g. Mujoco or Deepmind 3D lab)

Correctness: * The methodology largely seems correct to me. * For the empirical results, why not have an RL baseline for the Catch environment?

Clarity: * The paper is well written. I especially appreciate the notation and natural lead-up to the algorithm description in section 3. * The description of results on the Catch environment could use some more detail. Particularly, what exactly does the 3-step lookahead baseline do? Is it trained or does it simply perform a search based on pre-defined values?

Relation to Prior Work: Yes, in pretty good detail, including a table of features of different works.

Reproducibility: Yes

Additional Feedback: * Line 112: The authors mention that one intuition for predicting targets instead of the objective directly is to ensure that learning always moves the agent's predictions towards the target rather than away. What happens if the target itself jumps around haphazardly? How different would this be from the divergence effect that would like to be avoided? * * Re: reproducibility: it would be helpful to provide exact architecture details and release code publicly.


Review 2

Summary and Contributions: In this paper, the authors propose a new meta RL algorithm where the value prediction target is self-learned, i.e., generated by a trained prediction model. The value function learns to predict the self-generated value target at the inner loop of the meta RL algorithm, whereas at the outer loop, the value function learns to predict a canonical multi-step bootstrapped return. The target at the outer loss could be replaced by any standard RL update target. ------->>>> Post-rebuttal update The main results for this method are presented as synthesized learning curves over a set of Atari games (Figure 3 from the main paper) and the individual playing scores for each of the games are not provided from the appendix. It is good to see such scores in the updated version of this paper so that it is easier for the follow-up works to compare with this method.

Strengths: + The idea of formulating the inner loss for meta RL as learning from the objective discovered by its own is interesting and novel. Generally, defining the algorithm to self-discover its objective makes the learning algorithm moves one step closer towards developing automated machine intelligence compared to the conventional meta RL methods which greatly rely on expert's design choice such as the hyperparameter to perform learning-to-learn. + The authors present extensive experiment results to evaluate the proposed method. The proposed method has been evaluated on three task domains: a catch game to demonstrate the method could effectively learn bootstrapping, a 5-state random walk to demonstrate the method works in non-stationary environments, and ALE which is a large-scale RL testbed. In all the task domains, the proposed method achieves noticeable performance improvement over the compared baselines. + The authors propose a consistency loss for large-scale experiments, which is used to regularize the output of the target prediction model to be self-consistent over time. The consistency loss could bring significant performance improvement when testified on ALE, if its weight is properly set.

Weaknesses: - My main concern for the paper is that the proposed method has not been compared with many meta RL methods. It is good to have some meta RL baselines that performs hyperparameter optimization (e.g., [1]) or using a classic RL target different from the outer loop target return as the inner loop prediction target, etc. The effect of incorporating self-discovered object could have been evaluated more thoroughly. [1] Meta-Gradient Reinforcement Learning (Neurips 2018).

Correctness: The problem formulation is sound and can generally work for most deep RL problems.

Clarity: The paper is generally well-written and structured clearly and I enjoyed reading it. The method is formulated in a clear way and is easy to understand.

Relation to Prior Work: The authors present extensive literature review for meta RL and provide in-depth comparison with the existing literature.

Reproducibility: Yes

Additional Feedback:


Review 3

Summary and Contributions: This paper proposes to allow RL agents to learn their own objective during online interactions with their environment. First, a meta-learner takes the trajectory as input and meta-learns an update target. Then, a traditional TD-like update or policy-gradient update will be performed based on the learned target. Because the target is learned according to online trajectories, it can adapt to the changing context of learning over time. To speed up learning in complex environments, a simple heuristic is added to regularize the learning space of the target function towards targets that are self-consistent over time. This paper provides motivating examples to show their proposed method can address issues on bootstrapping, non-stationarity, and off-policy learning. In addition, the authors also conduct large-scale experiments on Atari to further evaluate their method. Contributions: 1. This paper makes a first step towards meta-learning update target in the context of meta-gradient RL. 2. The proposed method makes uses of online trajectories to allow the agent to learn its own online objective and thus learn how to learn increasingly effectively.

Strengths: 1. This paper tackles a very valuable problem of meta-learning knowledge from online trajectories. The idea of building a learnable update target for RL is interesting and novel. This paper is relevant to a broad range of researches on meta RL. 2. They provide motivating examples to validate their motivation of algorithm design. These experiments are well-designed to support the main claims.

Weaknesses: 1. I wonder if R2D2 or an extension version of a conventional model-free RL that supports POMDP will have similar performance with the proposed method in this paper. For example, what if we simply use a trajectory encoder to capture the true state through history and then add it to the value network and policy network as an additional input feature. Maybe this naïve method can also address tasks like “Catch” and “5-state Random Walk”. More experiments will be helpful to further understand the contribution of this paper. 2. The proposed approach has only comparable or slightly better performance than baseline method (IMPALA) on large-scale standard Atari benchmark, but it has a much more complex implementation than baseline method (a meta learning algorithm seems to be harder to train and use compared to a simple model-free method, and may be less stable in practice). In addition, some recent stronger baselines(e.g., R2D2, NGU, Agent57, and MuZero)on this dataset are not included.

Correctness: The claims and method are correct. The empirical methodology is sound.

Clarity: Overall, the paper is well-written. I only have a few suggestions. 1. It will be better if an algorithm sketch or pseudocode is provided. 2. Some notations are a little bit confusing. In Section 3.4 and 3.5, g and G represent learned update target and returns, respectively, but in Line 231, G changes to learned update target. 3. In Line 174, “When the pellet reaches the top row the episode terminates” should be “When the pellet reaches the bottom row the episode terminates”?

Relation to Prior Work: Yes.

Reproducibility: Yes

Additional Feedback: 1. In “5-state Random Walk”, what is the trajectory length used in g(\tau)? Is a trajectory within a single episode or crossing many episodes? 2. Are there any implementation codes? ========================================== post-rebuttal comment Thanks for the feedback. I increase my score to 6 and tend to accept this paper since the feedback has addressed my main concern. The authors provide additional experiments to show their methods can outperform a naïve extension of RL algorithm which uses trajectory encoder as additional input to value and policy network and in principle supports non-stationarity.

[Author Response · NeurIPS 2020]

**To Reviewer #2**: Thanks for recognising the novelty, clarity and promising results of our work.

**Reproducibility**: Please note that our supplemental material contains both the **complete architecture** of the agent and **detailed pseudo-code** for the algorithm. We'll add a reference to the appendix in the main text of the paper.

**Catch related**: We'll provide more details on the learning to bootstrap results. Note that the agent only sees a limited window of 3 time-steps, and hence cannot see all the way to the end of the episode to provide the full return. For your specific questions, 1) the 3-step look-ahead baseline is the theoretical optimal performance any agent can achieve without bootstrapping. If an agent performs better than this score, it shows that the algorithm has learned how to bootstrap. We'll provide detailed calculation of the theoretical optimal performance in the appendix; 2) We'll add an RL baseline for the Catch environment.

**Robustness of learning a target vs. learning a loss**: For any given $\eta$ the expected target $\mathbb{E}(g_\eta(\tau))$ is a well-defined function, even if the target jumps around "haphazardly" from state-to-state or step-to-step. Optimising a squared loss by SGD simply adjusts the value function $v_\theta(S)$ in a way that minimises MSE to the expected target. In contrast a learned loss could yield arbitrary dynamics (for example, moving *away* from a target) which might then diverge easily. We have observed this in our experiments. E.g., Figure 3b shows that learning a loss was more fragile and the agent failed to achieve good performance on large-scale experiments.

---

**To Reviewer #3:**  We appreciate your valuable comments! We will add a baseline for [1].

---

**To Reviewer #4**: Thanks for noting that our paper tackles a very valuable problem and that the idea is interesting & novel.

**Trajectory encoder as additional input to value and policy:**

The "5-state random walk" was designed to demonstrate FRODO can learn to handle non-stationarity. You are correct that a memory-based solution could in principle also deal with non-stationarity; however this is typically difficult when the time-scales are long (e.g. there are hundreds of steps between switches in the random walk example). Following your suggestion, we ran TD($\lambda$) using an LSTM to encode the history. Results are shown in the Figure 1 of the rebuttal, and the error metric is mean over 10 random seeds. FRODO performs better than this new baseline, suggesting it deals more effectively with non-stationarity.

"Catch" was used to show that FRODO can learn to bootstrap when given only a window of 3 time-steps into the future. Adding a trajectory history (e.g. LSTM) as input to the value function does not help with the issue of bootstrapping in Catch, and will result in the same theoretical upper bound as shown in the paper (a low score around 0.2).

Figure 1: Comparison of FRODO with trajectory encoded value prediction in "5-state random walk" environment (suggested by Reviewer #4).

**Comparisons to baselines:** Our goal is an algorithm that learns its own objective solely from experience of interacting with its environment. The comparison to IMPALA is to show that the proposed FRODO algorithm can learn its own objective to outperform a strong actor-critic baseline on challenging domains. Note that FRODO algorithm uses exactly the same outer loss as the IMPALA baseline. FRODO builds upon the IMPALA baseline and furthermore learns to outperform it eventually. Interestingly from our analysis, FRODO learns a very different objective from IMPALA. We'd like to point out that outperforming an well-established RL algorithm by learning discovering RL objective online is non-trivial and to our knowledge this work is the first one to achieve such strong performance.

As for stronger baselines in Atari such as NGU, Agent57, and MuZero, the main goal in the paper is not to achieve state-of-the-art performance in specific domain, but to introduce a general algorithm which learns its own objective online. Using IMPALA as outer loss is clean and simple to show the effectiveness of our proposed algorithm. As we've mentioned in the paper, any RL objective can be used as outer loss in the proposed FRODO algorithm. People can easily extend and apply our FRODO algorithms with more complicated RL algorithms like NGU and Agent57.

**Implementation and code:** Please note that we provided **full pseudocode** in Python format in the appendix.

**Trajectory length in $\tau$:** Trajectory $\tau$ is a truncated-length trajectory within a single episode. It is with length 10 in the "5-state random walk" experiment.

[Meta-Review · NeurIPS 2020]

The reviewers agreed that this is an interesting, novel, and well-executed contribution. Congratulation! I would like to bring up two issues that were raised in the discussion, and ask the authors to address them in their final version. - It would be good to add some insight on why the meta learned update actually performs better. - The improvement over IMPALA, although clearly evident, comes at the cost of a much more complex algorithm. This should at least be mentioned/discussed.